# Performance of volunteer community health workers in implementing home-fortification interventions in Bangladesh: A qualitative investigation

Haribondhu Sarma[1,2]*, Ishrat Jabeen[2], Sharmin Khan Luies[2], Md. Fakhar Uddin[2], Tahmeed Ahmed[2], Thomas J. Bossert[3], Cathy Banwell[1]

1 Research School of Population Health, College of Health and Medicine, The Australian National University, Acton, ACT, Australia, 2 Nutrition and Clinical Services Division, icddr,b, Mohakhali, Dhaka, Bangladesh, 3 The Harvard T.H. Chan School of Public Health, Boston, MA, United States of America

* haribondhu.sarma@anu.edu.au

**Data Availability Statement:** Data cannot be shared publicly as participants did not give consent for their transcripts to be shared in the public

## Abstract

### Introduction

BRAC, an international development organisation based in Bangladesh, uses female volunteer community health workers called Shasthya Shebika (SS), who receive small incentives to implement its home-fortification interventions at the community level. This paper examines the individual, community and BRAC work environment factors that exert an influence on the performance of SS.

### Methods

This qualitative study was conducted between the period of June 2014 to December 2016 as part of a larger evaluation of BRAC's home-fortification programme. Data were collected through in-depth interviews, focus group discussions, and key informant interviews and analysed thematically. The participants were SS and their supervisors working for BRAC, caregivers of children aged 6–59 months, husbands of SS, village doctors, and Upazila Health and Family Planning Officers.

### Results

Younger, better educated and more experienced SS with positive self-efficacy were perceived to have performed better than their peers. Social and community factors, such as community recognition of the SS's services, social and religious norms, family support, and household distance, also affected the performance of the SS. There were several challenges at the programme and organisational level that needed to be addressed, including appropriate recruitment, timely basic training and income-generation guidance for the SS.

domain. Data generated from icddr,b's research can be provided to interested researchers (Recipients) for secondary data analyses upon approval of a Data Licensing Application & Agreement by the icddr,b Data Centre Committee. Interested personnel is recommended to consult this with icddr,b IRB Coordinator Mr. M A Salam Khan (ssalamk@icddrb.org).

**Funding:** Research for this article was founded by the Children's Investment Fund Foundation (CIFF) of UK. The views, opinions, assumptions, or any other information set out in this article are solely those of the authors and should not be attributed to CIFF or any person connected with CIFF.

**Competing interests:** The authors declare that there is no competing interest regarding the publication of this article.

## Conclusion

BRAC's volunteer SS model faces challenges at individual, community, programme and organisational level. Importantly, BRAC's SS require a living wage to earn essential income for their family. Considering the current socio-cultural and economic context of Bangladesh, BRAC may need to revise the existing volunteer SS model to ensure that SS receive an adequate income so that they can devote themselves to implementing its home-fortification intervention.

## Introduction

The availability of adequately skilled health workers at the community level is important to ensure essential, life-saving interventions. Considering population density, South-East Asian and African regions, which bear the greatest burdens of preventable diseases, have the lowest density of health workers compared to the wealthier regions of the world [1]. The numbers of health workers and quality healthcare are positively correlated [2]. Several studies have suggested that countries with a shortage of health workforce suffer from impaired provision of important, life-saving interventions, such as immunisation, antenatal care for pregnant women, and nutrition services. This has resulted in a significantly higher level of disease burden, health inequalities, mortality, and morbidity [2–6]. Considering the shortage of healthcare-related human resources, the use of community health workers (CHWs) has become increasingly important in many low- and middle-income countries. The CHWs generally provide several services to communities, including culturally-sensitive health messages, social support, and they play a critical role to connect between community and broader health systems [7–11].

In many low-and -middle-income countries, there are two types of CHWs available: a paid CHW and a volunteer CHW [12]. The roles and responsibilities of the paid CHWs are clearly defined by the organisations that pay their salaries; they are generally accountable to their employers who influence to improve their performance. However, the roles and responsibilities of volunteer CHWs are flexible. They have flexible working hours and days so they can work when it suits them and the community. Their role is somewhat determined by the community, the CHWs themselves, or in consultation with organisation employing them [12]. The volunteer CHWs are generally unpaid; sometimes, they receive a modest financial incentive or other non-monetary gifts. This category of CHW is influenced by the spirit of volunteerism to serve people in their community without direct financial interest. Although both the models have advantages and disadvantages, performance of volunteer CHWs is of more concern compared to that of the paid CHWs because there is a shortage of women with required skills, a high dropout rate, and irregular home contact [12, 13].

In Bangladesh, there is a long history of using both paid and volunteer CHWs. The Ministry of Health and Family Welfare (MOHFW) recruited several paid CHWs, including Health Assistant, Family Welfare Assistant, and Community Healthcare Provider. These CHWs implement vertical and integrated health and nutrition interventions for the MOHFW. In contrast, several non-governmental organisations (NGOs), including BRAC (a development organisation, formerly known as Bangladesh Rural Advancement Committee) use female volunteer community health workers called Shasthya Shebika (SS) to implement its health and nutrition activities at the community level. BRAC recruited SS within the targeted

communities and provided them with basic training on a range of essential healthcare services. The SS working at the community level are directly supervised by Shasthya Kormi–who are a BRAC paid health staff. Each Shasthya Kormi supervised 8 to 10 SS. Their details, roles and responsibilities are described elsewhere [14]. The SS do not receive a salary or monthly stipend; they are provided with financial incentives to offer specific health services to communities and they make a small profit on the sale of BRAC's products, including basic medicines, selected health commodities, and nutrition products (e.g. Pushtikona–a BRAC product of micronutrient powder). Detailed information about the SS has been supplied in other papers [15,16].

### Role of the BRAC SS in home-fortification programme

In Bangladesh, the Maternal, Infant and Young Child Nutrition (MIYCN) Phase II is a large home-fortification programme implemented by BRAC between 2014 to 2018, targeted to reach about 15 million children aged 6–59 months in five years. The BRAC SS implemented the programme at the community level. The roles of SS in implementing MIYCN Programme Phase II include selling MNP sachets at the household level and providing counselling-education to the caregivers on the use of MNP. The detailed activities of the SS in the MIYCN programme are clarified elsewhere [16]. The programme aimed to reduce the prevalence of anaemia by 10% from the baseline by ensuring effective coverage of home-fortification with micronutrient powder (MNP). Under the MIYCN Programme, SS received basic training at the beginning of the programme and refresher training every month during implementation. The SS were also responsible for visiting households at regular intervals to counsel caregivers of children aged 6–59 months on MIYCN home-fortification with MNP. Each SS received a profit of around BDT 16 (1 BDT = 0.012 USD) for selling a box with 30 sachets of MNP product. Each SS also received an incentive of BDT 60 if she could sell three boxes or 90 sachets of MNP product to the caregivers. During initial evaluation of the activities of MIYCN Programme, the performance of SS has been identified to be very critical for the success of the programme.

The performance of community health workers has been influenced by several factors, including availability of CHW, productivity, competence, and responsiveness [17]. Previous literature identified a number of concerns around performance of the BRAC SS, including, high dropout rate, inactivity, irregular and infrequent home-visits [16,18–20]. Considering these concerns, this paper aimed to explore individual-level factors, BRAC work environment was including organisational and structural perspectives, and socio-cultural issues at the community level that influence the performance of the SS.

## Materials and methods

### Conceptual framework to assess performance of the BRAC SS

Our framework was developed based on a socioecological model in which individuals are embedded in a larger socioecological system. It described the characteristics of individuals and interactions between them and the environments that underlie health outcomes [21]. We adapted a framework developed by Gopalan *et al*. [22] which had been implemented in a similar low-income setting where BRAC SS are working, to assess their 'performance motivation'. It was also based on a literature review of the factors influencing the CHW performance [16,18–20] which identified 19 parameters under the four broad categories: individual, community, organisational, and programme-level. Individual-level characteristics include age, education, perceptions, and social norms of CHW. The environmental factors include community-level factors, organisational factors, and programme-level factors (Fig 1).

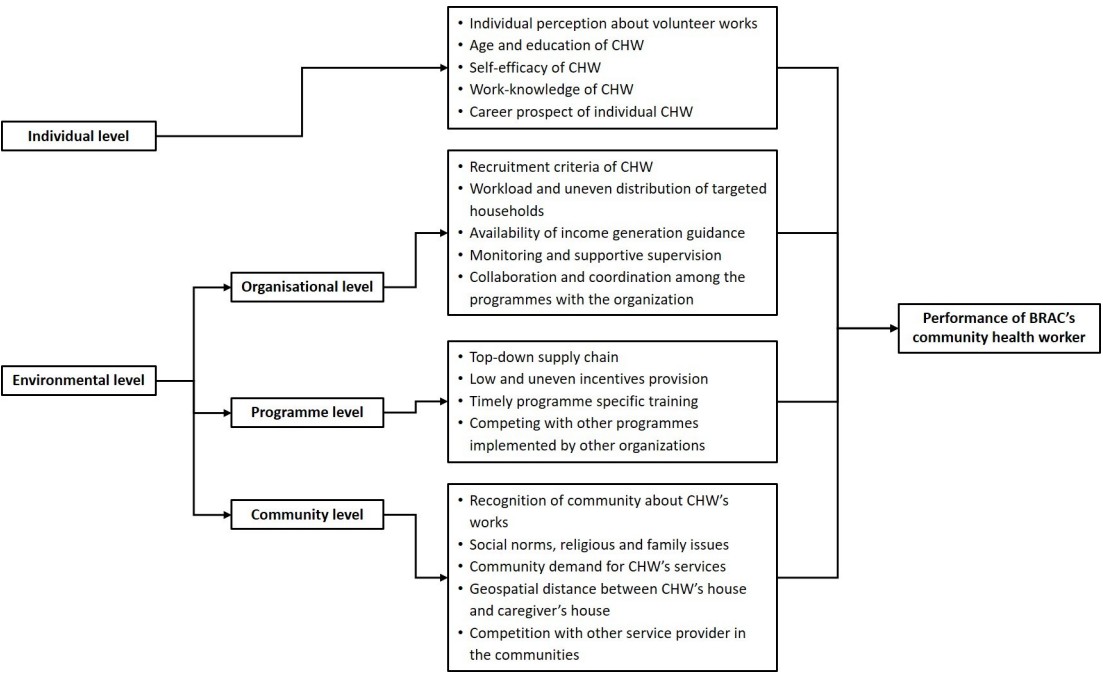

**Fig 1. Conceptual framework for performance of BRAC's community health workers in Bangladesh.**

## Study design

This qualitative study was conducted as part of a large concurrent evaluation of the MIYCN Home- fortification Programme during the first two years of the programme implementation. Details about the design of the concurrent evaluation have been explained in another paper [16]. We conducted a series of qualitative investigations during implementation of the MIYCN Programme between June 2014 and December 2016. The aim of this integrated approach was to provide flexibility to fill in gaps in the available information, to strengthen the validity of the individuals perceptions and to provide different perspectives on complex, contextual, and multi-dimensional phenomena around performance of BRAC's SS in home-fortification programme.

## Study sites

The MIYCN Home-fortification Programme was implemented in 164 rural sub-districts and 6 poor urban areas of Bangladesh. We conducted the study in 19 (11%) of the program's rural areas and 1 (16%) of the poor urban areas, providing a roughly similar proportion of each. When selecting study sites, we aimed for diversity across the sub-districts. For example, we selected study sub-districts based on the availability of SS in the sub-district, duration of home-fortification interventions (implementing more than one years or less), presence of other NGOs with home-fortification interventions in the sub-districts, vacancy in BRAC's staff positions (area with high vacancy vs. low vacancy), and the geographical settings of the sub-districts (e.g. hard-to-reach locations). However, there was little variation between the six urban areas where the fortification program was implemented.

## Data collection

An experienced team of three female and four male social scientists with postgraduate educational backgrounds in anthropology or sociology was involved from the beginning of the

investigations. The team had prior experience of up to 10 years in conducting qualitative research. The first author of this paper was the principal investigator (PI) of the evaluation and led all qualitative data collection as well as monitored and supervised the activities of other team members.

In order to generate in-depth, holistic information, to reach data saturation, and to perform data triangulation, we used multiple data collection techniques. We conducted face-to-face in-depth interviews with the SS and their husbands, as SS performance influenced by her husband's understanding and motivation about her works. We also conducted in-depth interview with caregivers of the target children who received services from the SS. We also conducted focus group discussions (FGD) with the SS and Shasthya Kormis (the paid supervisors of the SS) and key informant interviews with Programme Organizers, Field Organizers, Upazila Manager–Nutrition and District Supply Chain Officer from BRAC's MIYCN Programme and with Upazila Health and Family Planning Officers (UH&FPOs) from MOHFW who had authority to approve any health and nutrition programme at the sub-district level.

We identified the study participants purposively considering the aims and objectives of the study to gain a holistic understanding of the issues across different contexts. For example, we selected a wide range of SS and caregivers, e.g. best-performing SS, medium-performing SS, young or old SS in terms of their age; in the selection process. We also considered similar attributes of the caregivers, e.g. caregiver with younger child or older child, caregiver with malnourished or well-nourished child, caregiver as biological mother, or any family member other than the mother (e.g. grandmother), regular or irregular users of MNP and caregivers who never used MNP.

We developed semi-structured interview guidelines for conducting the interviews and FGDs. The interview guidelines for the SS mainly covered the barriers and facilitators that the SS faced while implementing the BRAC's Home-fortification Programme. The four main themes of the conceptual framework overlap in the interview and FGD guidelines. The individual and community-level factors were covered in interviews with SS and caregivers. The FGD with Shasthya Kormi included individual, community and programme level factors. Information relating to the organisation and programme was also collected through key informant interviews with BRAC Managers.

Before conducting interviews and FGDs, the respective interviewers built a rapport with the respondents and elder family members of the households. During rapport-building, the interviewers introduced themselves and explained the research projects, including aims, objectives, data collection, analysis and reporting procedures. All interviews were conducted in household settings and FGDs were conducted at local BRAC's offices. Before conducting interviews/ FGDs, the interviewers chose a quiet location, where only respondents and researchers were present. Interviews and FGDs took on an average, 1 to 1.5 hours and 2 to 2.5 hours respectively.

All the interviews and FGDs were recorded digitally with permission from the participants. Immediately after conducting each of the interviews and FGDs, the respective interviewers prepared a transcript with detailed notes of the interviews and FGDs. Then they shared the transcripts among the PI and other senior team members for their initial review and feedback. The process helped the team develop an understanding of the emerging themes and potential gaps in data collection, and created opportunity to address these gaps and clarifying the emerging issues in the subsequent interviews and FGDs.

During interviews with the SS, we collected a range of information, including their knowledge of the MIYCN Home-fortification Programme and perceptions about working as an SS; community-level issues (e.g. how the community recognises the SS's work, socio-cultural sensitivities to work as an SS), and their interaction with BRAC, including experience in receiving

a monitoring and supervisory visit. We conducted interviews with caregivers of the children of 6–59 months to understand their experiences of receiving services from the SS, how they interacted with the SS, and how effective the services were.

FGDs with Shasthya Kormi also covered a number of issues, including the experience of Shasthya Kormi working with the SS under her supervision, how Shasthya Kormi assessed the strengths and weaknesses of an SS, what the feedback mechanism was, how they interacted with both BRAC as an institution and the SS as community volunteers, and what the challenges were that the Shasthya Kormis observed regarding performance of the SS. From key informant interviews, we collected information about BRAC's policies about the SS model, BRAC as an organisation, and the MIYCN home-fortification programme, and dependence on the SS as community volunteers to implement the interventions at the community level. We also collected information about the organisational and programme-level barriers and opportunities that influence the performance of the SS.

## Data analysis

The conceptual framework of the performance of the SS of BRAC (Fig 1) guided the overall analysis in this paper. We used a thematic analysis based on the framework method proposed by Ritchie J *et al*. [23] described below. Since data for this paper were generated through a large evaluation conducted with multiple qualitative investigations, this analytical method provided clear steps to follow and generated a highly-structured output of summarised data [24, 25]. We used NVivo (Version 11) for managing and coding all data. After transcribing interviews, we spent a substantial amount of time becoming familiar with the transcripts by reading and rereading them and the interview notes. During reading and re-reading, we labelled the texts with codes. We followed both deductive and inductive approaches for coding data [26]. In the deductive approach, the conceptual framework for assessing performance of the SS guided us to formulate themes.

The first three authors coded data from the transcript and shared the results to identify the final code list. After coding the data, we developed an analytical framework by using a matrix table and categorising different codes into a broad theme. The matrix table enabled us to display all data in one frame, summarise data by each transcript and reduce the data, which were not relevant to the study objective. This exercise also helped us to triangulate the findings, generated from different data collection techniques: in-depth interview, FGD and key informant interview. We then developed descriptive and explanatory accounts, by summarising and synthesising the range and diversity of coded data [25]. In this step, we spent time interpreting the findings and identifying characteristics and patterns. Finally, we presented the findings within the outline of conceptual framework. We also used verbatim statements of the study participants to present some complex and critical findings.

## Ethical approval

The institutional review board of icddr,b which consisted of two committees: the Research Review Committee and the Ethical Review Committee approved this study. We used a written consent form for conducting interview and FGDs; prior to the data collection, we read out the consent form to the participants, responded to all queries from the participants, and then asked them to give consent.

## Results

### Background characteristics of study participants

A total of 186 participants (SS, Shasthya Kormi, Caregivers, etc.) participated in the study, with an average of 7 from each sub-district (range 1–10). Among the total study participants, 10 participants came from the urban study site. A total of 69 SS participated in this study through 14 in-depth interviews and 10 FGDs. We conducted two in-depth interviews with respective husbands of the SS. A total of 56 Shasthya Kormi participated in the study through 32 in-depth interviews and three FGDs. A total of 17 BRAC staff members from managerial level participated in the study through key informant interviews; four of them were District Managers–Nutrition, nine were Upazila Managers–Nutrition, and four were Supply Chain Officers. We also conducted two key informant interviews with UH&FPOs and one interview with a village doctor (non-formal health care provider). Additionally, we conducted 22 in-depth interviews with the caregivers of the children of 6–59 months (S1 Table).

Table 1 described background characteristics study participants. The mean age of the SS was 41 years, which ranged from 22 to 60 years. However, the Shasthya Kormis who were the

**Table 1. Background characteristics of study participants.**

| Background characteristics | Mean | Range | |
|---|---|---|---|
| | | Lowest | Highest |
| **Background characteristics of Shasthya Shebika (SS)** (N = 69) | | | |
| Mean age in years | 40.75 | 22 | 60 |
| Mean years of schooling | 4.3 | 0 | 15 |
| Mean years of experience as SS | 6.6 | <1 | 20 |
| **Background characteristics of Shasthya Kormi (SK)** (N = 56) | | | |
| Mean age in year | 31.9 | 21 | 43 |
| Mean years of schooling | 10 | 9 | 12 |
| Mean years of experience as SK | 5.6 | <1 | 12 |
| **Background characteristics of Programme Organiser (PO)** (N = 17) | | | |
| Mean age in year | 31.6 | 27 | 38 |
| Mean years of schooling | 16 | 12 | 17 |
| Mean years of experience as PO | 4.6 | <1 | 15 |
| **Background characteristics of Upazila (sub-district) Manager (UM)** (N = 9) | | | |
| Mean age in year | 38 | 33 | 48 |
| Mean years of schooling | 16 | 16 | 17 |
| Mean years of experience as UM | 8.9 | <1 | 12 |
| **Background characteristics of District Manager (DM)** (N = 4) | | | |
| Mean age in year | 38.3 | 29 | 42 |
| Mean years of schooling | 17 | 17 | 17 |
| Mean years of experience as DM | 7.6 | 5 | 12 |
| **Background characteristics of Supply Chain Officer (SCO)** (N = 4) | | | |
| Mean age in year | 40.3 | 38 | 42 |
| Mean years of schooling | 17 | 17 | 17 |
| Mean years of experience as SCO | 5 | 1 | 13 |
| **Background characteristics of caregivers** (N = 21) | | | |
| Mean age in year | 27.25 | 19 | 37 |
| Mean years of schooling | 8 | 5 | 15 |
| Mean years of experience as a caregiver of children U5 | 5 | 9 | 12 |
| Mean number of children of each caregiver | 2 | 1 | 5 |
| Mean age of U5 children of each caregivers | 23.8 | 9 | 60 |

supervisors of SS were much younger than the SS; their mean age was 32 years. An SS completed on average, four years of schooling, which ranged from never-attending school to 16 years of completed education. About 20% of the SS never went to school, and could only sign their names. However, about 30% of the SS who participated in the study completed secondary level of education. The Shasthya Kormis, had a mean of 13 years of schooling. On average, an SS worked with BRAC for about seven years, (range one month to 20 years); their supervisor's (Shasthya Kormi's) mean experience was six years. The mean age of caregivers who participated in this study was 27 years (range 19 to 37 years). On average, caregivers completed eight years of schooling, and all caregivers completed at least primary level of education; some of them had a postgraduate degree (Table 1).

## Performance of the SS of BRAC

The performance of the SS has been found to be associated with several issues at the individual, community, organisational and programme-levels. Table 2 provides examples of the main study themes and sub-themes and the number of participants (SS, Shasthya Kormi, Caregivers, etc.) mentioned about the themes and sub-themes. Table 3 provides quotations of the study participants against the sub-themes.

**Table 2. Number of participants mentioned each of the themes at different level for assessing the performance of BRAC Shasthya Shebika (SS).**

| Theme | Number of participants* |
|---|:---:|
| **At individual level** | |
| Perception about working as an SS influenced performance | 6 |
| Age and education of SS were important predictors, but not always | 9 |
| Self-efficacy–hesitation to visit all household | 14 |
| Work-related knowledge helped SS to better perform in the communities | 7 |
| Career prospect–working as a SS was a point of entry to future betterment | 6 |
| **At community level** | |
| Community members had mixed perceptions about SS | 8 |
| Social norms, religious issues, family support influenced the functions of SS | 14 |
| Community demand for services from SS critical to their performance | 20 |
| Geospatial distance was important: the further households were from the SS's house the fewer SS visits | 7 |
| SS have been struggling to compete with other service providers in the communities | 8 |
| **At organization level** | |
| Appropriate recruitment of SS is critical to their performance | 8 |
| Workload and uneven distribution of households created concerns for SS | 7 |
| Inadequate income-generation guidance for SS influenced SS motivation | 19 |
| Regular monitoring and supportive supervision are critical to performance of SS | 4 |
| Collaboration and coordination with other BRAC's programmes influenced performance of SS | 4 |
| **At programme level** | |
| A top-down supply chain was responsible for stock-shortages of BRAC supplies at the SS level | 6 |
| Low and uneven incentives demotivated SS | 18 |
| Timely programme-specific training improves SS performance | 8 |
| Competition with other programmes is challenging for the SS | 13 |

* Number of participants mentioned about the theme

**Individual level.** *Perception about working as an SS influenced performance.* The active and better-performing SS perceived her work as an opportunity to have a formally recognised identity in the community and to work with the most disadvantaged population in the community. Before joining BRAC as a SS, most were housewives with no paid employment (Table 3). They were mainly recruited from lower socio-economic families and did not have any other identity in their community other than housewife. Working as a SS gave them an affiliation with BRAC, a well-known organisation not only in Bangladesh but also globally. We did not observe such positive perceptions among all the SS in the study, particularly those who were inactive or irregular in working as SS or were a lower-performing SS.

*Age and education of SS were important predictors, but not always.* One of the major selection criteria of SS was their age, which was 20–35 years. However, there were a number of SS who were above 50 years of age because, in the BRAC's model, there was no procedure for terminating an SS from her work (Table 3). Recently recruited SS had to have at least 8 years of schooling (formal education). Earlier, SS had been recruited without considering their educational level. Consequently, many non-literate women or women with below primary-level education had been selected as SS. These categories of SS faced several challenges; they could not effectively receive the training as they were unable to read and write. They also faced difficulties in performing their regular activities, could not maintain their daily registration and the simple calculations that they needed for their work. Although age and education helped SS perform better, we observed there were some SS in this study who were non-literate and aged but performed better than young educated SS. According to the BRAC staff members, these types of SS were naturally talented women; they mostly had learned how to face challenges and how to convince others (Table 3).

**Self-efficacy** refers to an individual's belief in his or her capability to perform behaviours related to produce specific performance attainments [27]. In the BRAC home-fortification programme, the self-efficacy of an SS has been identified as an important factor associated with her performance. Some SS, irrespective of their age and educational levels, were not keen to visit all households; rather they visited selected households. The SS reported a number of reasons for this which denoted low self-efficacy to overcome the challenges in the communities. For example, SS avoided visiting households where household members belonged to a different religion (Table 3), or where they thought they might not get access. The SS avoided the households with older people who were likely to ask more questions and did not allow them to spend time with caregivers. They also avoided households with educated caregivers who asked more critical questions which they were unable to answer. However, SS felt comfortable to visit households containing a relative and even the caregivers also felt comfortable to discuss any health problem with a SS who is their relative (Table 3).

*Work-related knowledge helped SS to better perform in the communities.* SS with adequate knowledge about the services were able to deliver them effectively. In the home-fortification programme, there were several tasks, such as counselling mothers about home-fortification of foods with MNP, demonstrating how to mix the MNP with food, explaining the benefits and side-effects of MNP, and responding to the caregivers queries. We observed that the SS who had very clear understanding about home-fortification could clearly explain it and demonstrate it to the caregivers without any hesitation (Table 3). However, their knowledge was influenced by various circumstances, including whether they had received basic training and regular refresher training, or whether they had asked questions or sought help from the trainers and the supervisors while working at the communities. Some SS who participated in the study rarely asked any questions to their supervisors and left difficult tasks for their supervisors. The Shasthya Kormis of those SS needed to spend more time and effort with them to achieve their work-related targets.

**Table 3. Illustrative quotations from study participants against main themes and sub-themes.**

| Level/ Main-themes | Sub-themes | Quotations of study participants |
|---|---|---|
| Individual level | Perception about working as an SS influenced performance | *"When I have started working with BRAC as SS, people in my community knew me; they called me BRAC's kormi (health worker of BRAC), this give me a new identity–earlier I was known as 'omuker boui' (wife of SS husband's)" (SS during In-depth Interview)*<br>*"We generally do nothing at home. As we work here, we can pass our time. As we are working for mothers and children, the mortality rate has reduced in my community. I feel very happy. I get respect from the community. That's why I work with BRAC as SS." (SS during FGD)* |
| | Age and education of SS was important predictors, but not always | *". . . . the aged SS who are 50 years or older can't work properly, they don't know the names of the medicines, even they don't visit the households and are unable to counsel mothers or caregivers properly." (Shasthya Kormi during in-depth interview)*<br>*"We found a woman in one of our tuberculosis (TB) clinics; her husband was suffering from TB for a long time, . . . . she had a very good understanding about TB and its treatment as she has been nursing her husband for a long time; when we recruited her as SS in our TB programme, she was initially hesitant as she was older and non-literate. However, after 18 days of basic training, she improved a lot. She eventually worked in different programmes very successfully and was rewarded as best-performing SS couple of times." (Upazila Manager–Nutrition)*<br>*"The SS, with lower level of education, have less capacity to explain. It causes difficulties in selling MNP, as they can't answer or explain mothers on specific topic when mothers ask them any question. . ." (Programme Organiser during Key Informant Interview)* |
| | Self-efficacy–hesitation to visit all households | *". . . . I felt uncomfortable to visit a conservative Muslim family, they might refuse me because of my religion, I usually avoided those houses, or visit when Shasthya Kormi apa come to my areas." (Hindu SS during in-depth interview)*<br>*". . . she (SS) is my relative from my in-law side. That's why I am more interested to take her service . . .maybe I wouldn't be much interested if another person came" (A Caregiver during in-depth interview)* |
| | Work-related knowledge helped SS to better perform in the communities | *"Nobody can be compared with X Apa (mentioning the name of an SS) as she is the best one. She has received the basic training as a best participant; she also has received training on Pushtikona (a brand name of MNP) properly and is delivering the services for the last three years . . ." (Shasthya Kormi during in-depth interview)* |
| | Career prospect–working as SS was a point of entry to future betterment | *"BRAC trained her (SS) how to provide primary healthcare, dispense medicine and sell health products. . . recently she took a loan from BRAC and started her own business. . . . she runs a shop at her home, her income helped my family financially." (Husband of SS during in-depth interview)* |
| Community level | Community members had mixed perceptions about SS | *"BRAC gave us an apron when I wear it, I look like a doctor; during my household visit, people recognise me with this apron." (SS during FGD)*<br>*"I am not educated; so, they didn't want to rely on my words. Several times, I tried to convince them to buy Pushtikona; they refused; however, they would buy it if an educated person of the community recommend them to use it." (SS during in-depth interview)* |
| | Social norms, religious issues, family support influenced the functions of SS | *Many people think that NGO means a Christian organisation as fund for the NGOs are mainly coming from Christian-dominated countries; they might influence the community with their views and ideology. Therefore, they avoid the services from the SS of BRAC." (Upazila Manager–Nutrition during key informant interview)* |
| | Community demand for services from SS critical to their performance | *"Demand of Pushtikona need to be increased. SS felt difficulties to sell Pushtikona due to lack of demand among the community. But they don't need to do that in case of other product." (Programme Organizer* |
| | Geospatial distance was critical: households far from SS's house get less SS visit | *"It is true that SS faced difficult to visit houses are far distance from her houses. This became worsen during rainy season. It would be the reason of irregular visit to the distant households." (District Manager–Nutrition during key informant interview)* |
| | SS have been struggling to compete with other service providers in the communities | *"We are from a family with low socio-economic background, and we do not have higher education. . . ... how do people trust us more than the others? We are always struggling to compete with a village doctor as they are better-educated than I, and they have a very good family background." (SS during FGD)* |

*(Continued)*

**Table 3.** (Continued)

| Level/ Main-themes | Sub-themes | Quotations of study participants |
|---|---|---|
| Organizational level | Appropriate recruitment of SS is critical to their performance | *"The selection criteria have some weaknesses. Actually, we do not always find SS according to the selection criteria. For this reason, often we recruit women who are a bit aged. If we search people according to the selection criteria, they do not agree to work. They demand more benefit against their service, which we are unable to provide." (Upazila Manager–Nutrition during key informant interview)* |
| | Workload and uneven distribution of households created concerns for SS | *"When I joined as a manager in 2015, there was 540 SS and 64 Shasthya Kormis in my area. In 2016, we reduced the number of SS to 294 and Shasthya Kormis to 32. In February 2017, again the number of SS was reduced to 190 and Shasthya Kormis to 19 while the target population remained the same. This increased the targeted households for an SS as she has to cover the area of another SS who dropped out. Initially, their targeted area was close to their residence, but now they have to move to distant places that involves more transport cost compared to their travel allowance. Thus, they became less interested in household visits." (Upazila Manager–Nutrition during key informant interview)* |
| | Inadequate income-generation guidance for SS | *"In the intervention area, the SS were rewarded with one box of Pushtikona if they could sell six boxes. It was a big reward for them. They (SS) thought that as much as they could sell, they would be benefited. For this reason, they increased their home-visit to seek the eligible children." (About initial effect of a business model, a District Manager-Nutrition in an area where business model was piloted said during key informant interview)* |
| | Regular monitoring and supportive supervision critical to performance of SS | *"We just help them by giving advices. They counsel the mothers; they convince them; they sell products. They are supposed to go with us when we visit the household. Suppose, there is a mother with seven months old kid and the SS demonstrates the mother how to feed Pushtikona to the child. During such demonstrations, we provide her with feedback if we find anything to improve." (Shasthya Kormi during FGD)*<br>*"We have seen in the fields that many mothers are currently using MNP, it proves that SS are able to convince mothers to use home-fortification with MNP; we also found some SS are unable to ensure that the children in their areas are fed MNP-mixed foods. In that case, a Shasthya Kormi might not be able to explain well to the SS about the demonstration. If we need to fill this gap, we send a PO to that area so that the SS would not face the problem." (Upazila Manager–Nutrition during key informant interview)* |
| | Collaboration and coordination with other BRAC programmes influenced performance of SS | *"Now, in a regular basis, we meet with the staff members of other programmes of BRAC, namely Dabi, Progoti, or Shikkha. They were informed about our home-fortification programme. They also have different types of village forums in different areas where they educate (inform) the group of community members, such as teachers and mothers. So, they could easily inform those community people about our programme besides their own." (Programme Organizer during key informant interview)* |
| Programme level | Top-down supply chain was responsible for stock-out of BRAC commodities at the SS level | *"Sometimes, we don't receive adequate amount of Pushtikona sachets; maybe the manufacturer doesn't supply according to our demand. I have to distribute the Pushtikona every month here. It has been seen that if I place a demand for 12,000 Pushtikona sachets, I only received 2,400–2,800, which is very depriving. Only once I received eight thousand Pushtikona sachets that was the highest amount I ever have received" (Upazila Manager–Nutrition during key informant interview)* |
| | Low and uneven incentives demotivated SS to work better at the community level | *"They might be less interested to sell Pushtikona as it is less profitable. The SS often says that we buy calcium at 10 taka and sell it at 15 taka; but you asked to sell Pushtikona at 75 taka. Buying at 56 taka, if we sell it at 75 taka, how much money would we get?" (Shasthya Kormi during FGD)* |
| | Timely receiving programme-specific training helped SS perform better | *"Training for the SS is a matter of time. We can hire SS but it is difficult to find such people who are willing to work. So, if any SS has dropped out, it takes couple of months to recruit, then couple of months to train the newly recruited SS." (District Manager–Nutrition during key informant interview)* |
| | Competing with other programmes was challenging for SS | *"In my area, World Vision Bangladesh distributes MNP free of charge to the caregivers of my targeted children whereas we the BRAC workers are selling it; so, why a caregiver would buy it from us as they are getting it free of cost from others. Moreover, for their free distribution, the caregivers who purchased MNP from us earlier are not trusting us anymore as they thought we cheated them by selling this product; they were supposed to get free from us as well." (SS during in-depth interview)* |

*Career prospects–working as a SS was a point of entry to future betterment.* Although the SS perform voluntary jobs and receive minimal financial incentives, the SS felt that this work may create an opportunity for getting a better job in future. The experiences gained by working as an SS may be valuable in terms of increasing self-confidence and life-skills. The SS reported that, when they were at home, they did not have opportunity to mix with people other than their family members. During community visits, they communicated with many people; they counselled them about healthy behaviours; they sold their products to the caregivers of children, and they maintained a business by themselves. Through their work, the SS become familiar to the community. Several SS were elected members of the local council based on their popularity. The experience of working as an SS was considered an important achievement that they could use when applying for other jobs relating to community-based functions. Some SS also used this experience to run a business (Table 3).

**Community level.** *Community members had mixed perceptions about SS.* As the SS sell health products and provide advice on health issues in the communities, they are recognised as a local doctor; people called them *dakterni* (female doctor) (Table 3). The SS reported that they appreciated this identity. However, sometimes community members did not accept the SS, as most of them were recruited from socially and economically-disadvantaged families. When their socio-economic status was lower than that of the visited households, household members refused her services. A SS is often ignored (Table 3) if their educational level is lower than that of the caregiver of the child.

*Social norms, religious issues, family support, influenced the functions of SS.* In Bangladesh, religious beliefs influence the social environment while individual lifestyles also determine the roles of men and women in the community. In some communities, religious leaders consider NGOs' activities to be non-Islamic and associate them with Christianity. They deter community members from being involved with NGO services (Table 3). Several participants from rural areas said that most of the people in the rural areas practised Islam and they were very sensitive about religious norms. They believed that Islam does not permit a woman to work outside the family. Due to these norms, some community members ignored the SS and discouraged other family members from receiving their services.

*Community demand for services from SS critical to their performance.* Our analysis revealed that when people in a community knew about the functions of the SS, they requested their services. In such a situation, the SS were comfortable with their work. Interviews with caregivers revealed that many were not aware of SS services until they received a home visit from them. There was very limited advertising by BRAC to raise community awareness about the SS's functions. However, in sub-districts where BRAC also used community mobilisation activities, including an informational meeting with community leaders about home-fortification of the MNP programme, caregivers had better understanding about Pushtikona (e.g., BRAC's MNP product) and the performance of the SS was better. Some caregivers were concerned about the side effects of Pushtikona and stopped using it if they saw their child having problems (Table 3).

*Geospatial distance was important: the further households were from the SS's house, the fewer SS visits.* In Bangladesh, many households were far away or hard-to-reach for an SS especially during the rainy season and bad weather (Table 3). We observed that BRAC allocated communities to SS without considering the locations of their homes. If the home of an SS was in the middle of the community, it was easier for the SS to reach all households. When the home of an SS was situated on the outer border of a community, it because difficult for the SS to reach households situated on the other side. Distant households were relatively unknown to the SS. In that case, the SS were less confident about whether they would be welcomed by the household head to provide services or to sell their products. The SS also reported that they

sometimes required local transport such as a rickshaw or a boat to reach some households but were not reimbursed for this cost by BRAC, and could not afford to spend their own money to visit these households.

*SS are struggling to compete with other service providers in the communities*. There were a number of other service providers, including village doctors, quack doctors, pharmacists, private practitioners, and CHWs of the Ministry of Health and other NGOs, providing health and nutrition services to the community and creating competition. A local dispensary is very common in Bangladesh along with a village doctor who is usually well trusted. Community members often prefer getting medicine or nutritional products from them rather than an SS (Table 3). In a community, if a village doctor sells MNP, the SS in that community is not able to achieve her MNP sale target.

**Organisational level.** *Appropriate recruitment of SS is critical to their performance*. BRAC recently changed the recruitment criteria of the SS by considering a new organisational strategy and the sustainability of the SS model. In the new recruitment criteria, BRAC considered two critical factors: age and education of the SS. BRAC's staff members at the sub-district level reported that they were struggling to recruit SS who had all the recommended criteria. Since socio-economic conditions of rural Bangladeshi people have improved during the last two decades, the availability of paid work has made it more difficult for the BRAC's local office to find volunteer workers. This has led BRAC personnel to compromise on the recruitment criteria (Table 3) and recruit SS who are not eligible to perform assigned tasks.

*Workload and uneven distribution of households created concerns for SS*. The allocation of households to an SS depended on several factors. At the very beginning of SS service delivery model, an SS was assigned for 250 to 300 households in rural areas; however, this number has changed over time to synchronise with new and upcoming programme needs. In urban areas, BRAC allocated 150 households to each SS. Since the SS dropout rate was high and recruiting an appropriate SS became difficult, the local BRAC office asked SS in nearby communities to provide services to those households who had lost their SS. This led remaining SS becoming overburdened with double the number of households. In addition, BRAC recently reduced their paid staff positions due to programmatic changes and financial constraints (Table 3) which resulted in an increased workload and targets for remaining staff that eventually impacted on their performance.

*Inadequate income-generation guidance for SS influenced motivation*. The overall philosophy of BRAC SS model was that it was based on voluntary work. The SS did not receive payment for their work. However, to motivate them, BRAC eventually introduced several earning mechanisms, including cash incentives for some services and allowed them make a profit selling BRAC's products. A strongly motivated skilled SS can earn BDT 2,500 to 5,000 in a month (1 BDT = 0.012 USD) but many others could not. Apart from basic training and a monthly refresher training, BRAC did not provide guidance to the SS about financial security. Furthermore, most SS reported that they had to provide a free sample to caregiver to try before buying. Since most caregivers were female and housewives (not a main earner in the household), they depended on their husband or the other main earner in the household to buy anything. In such a situation, a caregiver always asked the SS to sell the product on loan/credit to be repaid once she got money from her husband. According to the SS, distribution of free samples and sales on credit created financial pressures which demotivated them.

In several evaluations, concerns were raised about SS retention, dropouts and irregular home-visits. Considering these, BRAC shifted from a volunteer SS model to a business model to enable SS become involved in income generation. In this business model, BRAC provided support to SS to develop business skills (Table 3). They started with recruitment of quality SS who received training in the new model and how to apply these skills in real-field settings. Our

interviews showed that SS had a limited understanding of their clients or records keeping. According to managerial staff, BRAC is planning to address income-generation for SS.

*Regular monitoring and supportive supervision are critical to performance of SS.* BRAC has a cascade system of monitoring and supervision with Shasthya Kormis, to Field Organizers and Programme Organizers who are paid to monitor and supervised the SS through regular field visits (Table 3). Supervisors played an important role in motivating the SS by providing advice on excellent service and on how to counsel mothers while visiting the field. The supervisors also identified who performed well and who did not; this was done by investigating targeted households. We observed a high turnover or dropout among the supervisory-level staff, particularly among Field Organizers and Programme Organizers due to a low and uneven salary structure for BRAC staff members at the sub-district level compared to other NGOs working in the regions (Table 3). Unavailability of supervisory-level staff members created a huge constraint on providing regular monitoring of and supportive supervision to SS.

*Collaboration and coordination with other BRAC programmes influenced SS performance.* BRAC has many programmes running in an area at one time. Staff members of one programme were often unaware of other programmes. This resulted in negative impressions at the community level among the caregivers. Previously, BRAC did not organise coordination meetings but after receiving evidence from previous evaluations, BRAC started coordination meetings across all ongoing programmes. BRAC established internal coordination among the MIYCN staff and other programme officials. These coordination meetings helped in building an effective promotion of MNP (Table 3).

**Programme level.** *A top-down supply chain was responsible for shortages of BRAC supplies at the SS level.* Initially, demand-notes and requisition of commodities had been prepared at the sub-district to the district level, then to the national level. In this process, community-level staff members were not consulted. Often, these requisitions did not reflect the actual demand and resulted in shortfalls in products which created challenges for the performance of SS at the community level. If the SS did not have adequate supplies of products for providing their services, they lose clients' trust who then refused products or services from the SS. During our initial qualitative data collection, we found supplies of MNP did not match community demand (Table 3). There were also issues at the manufacturer level; they could not always meet increased demand for products.

*Low and uneven incentives demotivated SS.* The performance of non-paid SS was reliant on incentives; without them, they became demotivated and inactive. The incentives have been reduced for many activities performed by the SS. Earlier, if an SS identified a pregnant woman from the community, she received 50 BDT; if she attended a delivery and early helped initiate breastfeeding, she received BDT 150. At present, the SS receives 20 BDT for identifying a pregnant woman; if she attends a delivery, she receives 10 BDT only. Due to such reductions, many SS do not want to perform the activity.

The SS received higher incentives from a nutrition programme which was phased out recently. However, other ongoing programmes were not interested continuing with these high incentives so that the SS became less interested in performing nutrition-related activities. As SS were used to receiving higher incentives over the past 4–5 years, they did not want to do the same work for no incentives. During the initial period of the MIYCN Programme, there was no incentive included in the home-fortification intervention; for that reason, the SS were reluctant to provide services or sell product (Table 3). Later, BRAC introduced incentives under the MIYCN Programme. If an SS can sell six boxes of Pushtikona, she receives 50 BDT as an incentive. Moreover, an SS receives 150 BDT when she ensures a child in her area is fed 120 Pushtikona sachets within 12 months with the condition of ensuring 60 sachets were sold

within six months. After introducing these incentives in the MIYCN programme, SS sales of MPN increased.

*Timely programme-specific training improves SS's performance.* Generally, BRAC provided basic training through the training division of BRAC when they started a new programme with SS. The training department of BRAC organised such training in batches with each batch containing 20 participants. The training department of BRAC is a separate department from the MIYCN programme department. They only organised training for newly recruited SS when they had 20 SS to fulfil a batch. Consequently, when BRAC field office recruited a new SS to fill a vacancy, they could not provide basic training to that SS immediately. Usually, a SS waits about four to six months to get the basic training even though they have started work (Table 3). This creates additional challenges for untrained SS which influenced to their performance in home-fortification implementation as they were unable to respond to questions from community members, which undermined their credibility. There were some other challenges in organising timely training, including delays in allocating a training budget.

*Competition with other programmes is challenging for the SS.* We observed that multiple organisations in the same community have provided interventions of home-fortification with MNP. In that situation, differences in the programme modalities among the organisations affected their workers at the community level. BRAC's MIYCN Programme followed a market-based approach–where SS purchased MNP from BRAC's local office and sold it to the caregivers of her community with a profit margin. Other organisations working in the same communities followed a free-distribution model by providing MNP to the caregiver free of charge and they implemented a piloted intervention. Having two programmes created confusion among the community members and created a difficult situation for the SS (Table 3). Our analysis revealed that an NGO was freely distributing MNP in one sub-districts of a BRAC community; therefore, BRAC SS stopped selling Pushtikona in these communities.

## Discussion

The BRAC's SS model is one of the largest health service delivery networks in low-income settings. Currently, BRAC's volunteer CHW model is in use in eight low- and middle-income countries and improve the health and nutrition of a huge number of underprivileged people [28]. In Bangladesh, despite a significant reduction in the number of BRAC CHW, it has the largest CHW network in the country. Currently, BRAC CHW contributes an important role to achieving the country's health and nutrition targets. In this context, understanding the performance of BRAC's SS has policy relevance. Our qualitative analysis suggested that the performance of their SS depended on several factors operating across individual, community, organisational, and programme levels.

As a key characteristic of BRAC's SS model is volunteerism, our analysis suggests that getting an adequate number of volunteers at the community level was a challenge for BRAC local offices. The issues around high dropout of the SS mostly related to no or insufficient earning options in the SS model [19, 20]. This is probably influenced by improvements in the rural economy of Bangladesh during the last couple of decades [29, 30] and the increased availability of paid jobs in agriculture and non-agriculture sectors [31]. Poorer rural women now have alternative sources of income available. Considering this changing socio-cultural and economic context, BRAC is moving towards a business model by providing SS with additional skills and guidance to increase their income. Additionally, BRAC should consider task-specific incentives to motivate SS to implement a particular intervention. Previous studies have suggested that financial incentives or reward were necessary to retain and maintain the engagement and motivation of volunteer CHWs [29, 30].

Age, education, and work-related knowledge of the SS are important for health workers [10]. Working as a SS is laborious, as they need to walk across the community carrying a bag of BRAC products (MNP and other BRAC products). It is evident that a young and energetic health worker can manage this task more easily than an aged health worker [10, 11]. Nowadays, a formal education is essential for a CHW to improve performance in communities. The CHW with a higher education has better work-related knowledge and improved performance [10, 21, 32, 33]. Moreover, volunteer and paid workers require enough education to use supporting technologies (e.g. digital register on a smartphone/tablet) and to effectively receive training about new programmes.

As reported in previous studies [10, 15], there were several community-level issues associated with the performance of the BRAC SS including recognition by, and demand for, SS's services in the community, the distance between SS's house and caregivers' house, and SS's ability to compete with other health service providers in the communities. Previous studies reported that community acceptance of the CHWs work depended on local community leaders [34] and that the distance between a CHW house and targeted communities influenced their performance [35, 36].

Many studies have found that socio-cultural norms and gender roles influence the performance of female CHWs [35, 37–40]. Our findings revealed that social norms, religious beliefs, gender role, and family supports were influential. It is difficult for CHWs to improve performance; they are do not fit with the socio-cultural contexts in the communities. It is recognised that female volunteer CHWs in Bangladesh are facing cultural and social challenges that restrict their work and create barriers to doing their jobs effectively [41, 42]. It implies that to improve SS performance, several initiatives are required including sensitising local community leaders, raising awareness in the community about their work, reallocating targeted households to reduce travel distance and establishing better coordination and collaboration among service providers available in the communities. CHWs themselves cannot perform these initiatives; they need support from the programme and organisational levels.

Issues identified at the organisational and programme levels overlap. To improve the performance the BRAC SS a combined and harmonised approach is required between the organisational and programme level. BRAC should put more work into predicting its future programme needs, then recruit SS considering critical criteria and provide timely training on how to provide services and address the challenges in the communities. Previous studies have found that volunteer CHW who do not have adequate training are less effective and are unlikely to achieve programme outcomes [35, 43].

BRAC needs to provide adequate support to SS through regular supervision and monitoring of their responsibilities, to provide opportunities to rectify misunderstandings and enhance their skills. Supportive supervision often requires on the job training [44, 45]. We also observed several constraints at the sub-district level of BRAC, including, a high dropout rate among supervisory-level staff members, staff members' concerns about salaries and benefits and coordination among the BRAC's programmes. BRAC should review existing pay structures for the supervisory-level staff members at the sub-district level and make these competitive with other NGOs and similar types of organisations.

## Strengths and limitations of the study

We conducted this study as part of a larger evaluation, which considered a range of qualitative data collected from multiple sources. Multiple interview data-collection sources allowed us to triangulate our findings, which ultimately improved the robustness of our conclusions. As is common in qualitative studies, we selected study participants purposively to ensure depth

rather than breadth of the evidence. Our findings are based on individual subjective perceptions. However, we ensured the robustness of these data by matching and crosschecking with data from different sources. This approach allowed us to present our findings, comprehensively and holistically. We only collected data on the BRAC SS who are volunteer CHWs. The performance of a CHW is a context-specific issue and may not be limited to individual, community, organisation and programme levels; factors at the policy and beneficiary levels might influence the performance of CHWs.

## Conclusion

There are several layers of barriers associated with the performance of BRAC's SS. At the individual level, they are age, educational status, self-efficacy and programme related knowledge of SS. At the community level, social and religious norms and community knowledge about SS services influenced SS performance. SS performance was also affected by the programme and organisational level factors including appropriate recruitment of SS, timely programme specific training, and regular monitoring and supervision. The availability of income generation guidance is critical for BRAC SS. Considering the current socio-cultural and economic contexts of Bangladesh, it is important to revisit the BRAC SS model as true volunteerism among SS will no longer work. Comprehensive income-generation guidance for the CHWs might help sustain this model in the long run. Ensuring community support and addressing organisational and programme-level constraints would support the BRAC's CHWs to work more effectively at the community level. Moreover, BRAC could collaborate with WHO to strengthen its CHW model by implementing WHO guidelines to optimise CHW based programmes and financing decisions to support human capital and health workforce development [46].

## Supporting information

**S1 Checklist.**
(PDF)

**S1 Table. An additional table to describe number of interviews conducted under each data collection techniques.**
(DOCX)

## Acknowledgments

We would like to acknowledge GAIN for technical support and BRAC for implementing the programme. We would also like to acknowledge Dr Catherine Harbour for her supports in accomplishing this evaluation. We thank all of the respondents for participating in this study and sharing their experiences in working with BRAC and providing valuable comments on it. We also thank all the study team members who are not included in the authors by-line but contributed during data collection, data management, and initial data analysis. icddr,b is also grateful to the Governments of Bangladesh, Canada, Sweden, and the UK for providing core/unrestricted support.

## Author Contributions

**Conceptualization:** Haribondhu Sarma.

**Data curation:** Haribondhu Sarma.

**Formal analysis:** Haribondhu Sarma, Ishrat Jabeen, Sharmin Khan Luies, Md. Fakhar Uddin, Cathy Banwell.

**Funding acquisition:** Haribondhu Sarma.

**Investigation:** Haribondhu Sarma.

**Methodology:** Haribondhu Sarma, Cathy Banwell.

**Project administration:** Haribondhu Sarma, Ishrat Jabeen, Md. Fakhar Uddin, Tahmeed Ahmed.

**Resources:** Haribondhu Sarma.

**Software:** Haribondhu Sarma, Ishrat Jabeen, Sharmin Khan Luies, Md. Fakhar Uddin.

**Supervision:** Haribondhu Sarma, Tahmeed Ahmed, Thomas J. Bossert, Cathy Banwell.

**Validation:** Haribondhu Sarma.

**Writing – original draft:** Haribondhu Sarma.

**Writing – review & editing:** Haribondhu Sarma, Tahmeed Ahmed, Thomas J. Bossert, Cathy Banwell.

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
