## [Decision Letter · Decision Letter 0]

16 Dec 2019

PONE-D-19-18797

Performance of Volunteer Community Health Workers in Implementing Home-fortification Interventions in Bangladesh: A Qualitative Investigation

PLOS ONE

Dear Mr. Sarma,

Thank you for submitting your manuscript to PLOS ONE. After careful consideration, we feel that it has merit but does not fully meet PLOS ONE’s publication criteria as it currently stands. Therefore, we invite you to submit a revised version of the manuscript that addresses the points raised during the review process.

We would appreciate receiving your revised manuscript by Jan 30 2020 11:59PM. To enhance the reproducibility of your results, we recommend that if applicable you deposit your laboratory protocols in protocols.io, where a protocol can be assigned its own identifier (DOI) such that it can be cited independently in the future. For instructions see: http://journals.plos.org/plosone/s/submission-guidelines#loc-laboratory-protocols

We look forward to receiving your revised manuscript.

Kind regards,

Jai K Das

Academic Editor

PLOS ONE

Journal Requirements:

**When submitting your revision, we need you to address these additional requirements:**

**Please ensure that your manuscript meets PLOS ONE's style requirements, including those for file naming. The PLOS ONE style templates can be found at http://www.plosone.org/attachments/PLOSOne_formatting_sample_main_body.pdf and http://www.plosone.org/attachments/PLOSOne_formatting_sample_title_authors_affiliations.pdf**

Reviewers' comments:

Reviewer's Responses to Questions

**Comments to the Author**

1. Is the manuscript technically sound, and do the data support the conclusions?

Reviewer #1: Yes

Reviewer #2: Yes

2. Has the statistical analysis been performed appropriately and rigorously? 

Reviewer #1: N/A

Reviewer #2: N/A

3. Have the authors made all data underlying the findings in their manuscript fully available?

Reviewer #1: No

Reviewer #2: Yes

4. Is the manuscript presented in an intelligible fashion and written in standard English?

Reviewer #1: Yes

Reviewer #2: No

5. Review Comments to the Author

Reviewer #1: General Comments

The manuscript provides a meaningful contribution to our understanding of factors affecting the motivation of Community Based Volunteers who have been incentivized. It is especially important for practitioners in Low and Middle Income Countries, where Community Based Volunteers deliver critical health services to a large proportion of the population mostly living in Rural areas.

Overall, the authors have attempted to provide clear descriptions and adequate information to understand the topic at hand. Their application of the framework analysis to arrive at the result was sound. However their discussion and conclusion sections need to be redone as they do not adequately capture the results and existing literature in the area of study. Additionally the authors should consider reviewing the grammar of their manuscript as it greatly affects the reading experience.

Specific sections

Abstract

1. Introduction: Based on the contents of the paper, I don’t think that the paper has evaluated SS performance. It has however evaluated the factors affecting the performance of the Shathya Shebika. I think it would be important to mention that they are voluntary workers but are paid a small incentive in the introduction.

2. Methods: The first sentence states that the study was qualitative in nature. The authors ought to combine the message in the next sentence to avoid repetition e.g “ This was a qualitative study conducted between June 2014- December 2016 as part of a larger evaluation of BRAC’s home fortification programme. Data was generated through in depth interviews, focus group discussions and key informant interviews…….”

3. Conclusion: The first sentence does not make sense. The authors can rephrase the sentence to read, “ BRAC’s volunteer SS model faces challenges at individual, community, programme and organizational level.”

Introduction

1. Line 88-90: Is the MIYCN ongoing or has it stopped this is quite unclear in the introduction section.

2. Given that the paper is discussing the SS performance in the MIYCN and factors affecting it, it would be informative for the authors to list some of their functions without necessarily going into detail in Lines 91& 92.

3. Line 99: I would suggest using another term instead of profit, as what they are receiving is not a profit rather a proportion of what they make selling the MNP.

4. Based on the descriptions provided in the subsequent sections of the paper, the authors should consider the aim/ scope of their paper to be an exploration of the factors affecting performance only and not an analysis of their performance.

Methods

1. The authors should consider revising the entire methods section. Despite providing critical information within the section, information that should be provided simultaneously is often disjointed. They should possibly consider the use of COREQ guidelines when writing this section to promote more cohesiveness and enhance the reader’s experience.

2. The authors state that they conducted interviews and focus group discussions but do not state the sample aside from those which were done with caregivers. This is critical information that they have listed under the results section. They should consider providing it here.

3. What was the rationale behind interviewing the husbands of SS?

4. Line 183: Change to FGD rather than GFD and quiet instead of quite.

5. Line 178: Built rather than build.

6. Lines 194- 213: This section provides a description of the data that was collected from each of the target populations. The authors could give information on the nature of the data they collected when they are explaining the conceptual framework and state that they developed interview guides based on it. Otherwise it feels like a repetition to state at the beginning of the section what guided the study then wait and describe it much later on.

7. Lines 213-214: This sentence should appear after Line 184, where the authors describe the actual data collection procedures.

Results

1. It is my understanding that the authors were trying to quantify the themes they came across in the data they analyzed and presented this as Table 2. However I am not sure that this adds value to their findings; For instance they state that in total 180 interviews were conducted, but only 20 participant mention community demand for services from SS as critical to performance. As such does this make it an important finding given that it was mentioned by few people?

Discussion

1. The authors have not provided a robust discussion that situates their work and findings within the literature that exists on the area of factors affecting the performance of community based volunteers.

Conclusion

1. The authors say that one of the limitations of their studies is that they only collected data on the SS who are volunteers. How can this be a limitation if the goal of the study is to evaluate the factors affecting SS performance and not that of paid community health workers?

2. Additionally given that their goal was to have an in depth understanding rather than be able to generalize findings. I don’t think they can state that the lack of generalizability of findings is a limitation of their study.

Reviewer #2: Overall Comments:

This is a well-designed and comprehensive qualitative evaluation study of an important area. There is a significant need for proper copy editing of the manuscript, as it is hard to follow in several places. Additional efforts to improve organization will aid with reduction of repetition. Issues with grammar and prose are pervasive, kindly review and edit throughout.

It is advisable that authors consider organizing their results by the different modalities of data collection, to ensure that they have captured key quotations and themes arising from all stakeholders involved in the process. Alternatively, highlighting more effectively/adding quotations throughout the results section, instead of restricting these to Table 3 will remedy this issue.

The role of Shastya Kormis is not clearly defined in the introduction and should be highlighted, particularly in reference to their involvement in provision of supportive supervision.

Consider citing the socioecological framework as your main theoretical framework.

A recent report by the WHO regarding the role of CHWs in LMICs should be cited, as it substantiates the need for payment and adequate monetary reimbursement to frontline workers. We have seen similar findings from studies conducted in geographically similar regional contexts.

The discussion is rather thin, and can benefit from a review of the literature pertaining to FLW performance and motivation. This is an important and timely issue and the overall findings in the study pertaining to adequate compensation for CHWs should be emphasized and highlighted with support from the existing body of literature in this area.

Introduction:

Lines 61-63:

- Statement is unclear

- Please highlight the roles and responsibilities of Shastya Kormis, their place in the overall BRAC organizational structure and compensation in relation to Shastya Sebikas

Line 106:

- “The performance of community health workers can be measured by several factors including availability, productivity, competence and responsiveness”

o Availability of what?

Materials and methods:

Lines 119-122:

- Sentence is unclear, please consider reframing

Lines 142-147:

- Can you comment on what point in the overall program this study was conducted? What was the duration between baseline and the evaluation?

Line 159:

- You mean to say “saturation”, in qualitative work, multiple data collection techniques are used to reach saturation of themes. This also has implications for your sample size.

Lines 159-176:

- This entire section is very hard to follow.

- Several points need to be addressed:

o How many interviews, FGDs and IDIs did you conduct and with which group

o Kindly reorganize this section by either – type of data collection method i.e. FGD, IDI, KII, or by the target respondent.

Lines 194-212:

- Same comments as above, this entire section is unclear and requires reorganization. For ease of organization, consider breaking the section up into sub-headings by type of data collection activity (i.e. FGD, IDI, KII) or target respondent.

Data Analysis:

Line 219:

- Your cited reference does not match your reference

Overall: did you engage in memoing during the data analysis process?

Lines 233-234:

- You mention triangulation of findings, can you clarify if you are referring primarily to the qualitative data or also the quantitative evaluation data? You mention triangulation, however it is unclear based on your results section, what the outcomes of triangulation were.

Results:

Lines 252-262

- This entire section is very hard to follow, same comment as above, please consider adding sub-headings and reorganizing to improve flow

Lines 287-289:

- This sentence is unclear, please rephrase

Lines 300-305:

- Consider adding a quotation here

Table 3:

- Age and education of SS

o Please add a quotation regarding education

o Also clarify whether you are referring to formal education or information/work related training

- Individual level factors:

o “self-efficacy – hesitation to visit all households”

This is unclear, what does self-efficacy have to do with this?

- Programme level factors:

o “Timely receiving programme….”

It is unclear what you mean to say in this section, additionally, the quotation you have cited does not match this factor

o “Competing with other programmes…”

This is an important factor, you may consider addressing it elsewhere in the manuscript as well. Why is BRAC selling MMPs in the same communities where other programs are ongoing?

Same comment for section on lines 420-421

• Why does this overlap exist and in how many communities was this seen?

- Lines 337-349

o Can you address issues of monetary compensation and/or performance in relation to improved training and performance of CHWs

- Lines 378-382

o Consider adding a quotation here to augment your findings

- Lines 479-481

o Why was a high turnover seen among supervisory level staff? Did your qualitative findings provide any additional insights as to what was driving this phenomenon? What were the implications of this for CHW performance?

- Lines 575-578

o Sentence is unclear – please rephrase

- Lines 583-585

o Sentence is unclear – please rephrase

- I am unclear on what the overall results from your FGDs and KIIs were, can you provide a synthesis of these in the results section.

- Can you briefly comment on the quality of data generated through the different modalities, i.e. in-depth interviews versus FGDs and whether this impacted the data you generated in any way?

Line 619-621:

- This sentence is unclear – please rephrase

Overall:

- Can you comment on selection bias and how you addressed this in your sampling strategy?

- Can you comment on any inclusion/exclusion criteria you kept in mind when sampling?

- You mention that you interviewed caregivers of beneficiary children, however I am unclear on what your key findings were from these interviews, and how they influenced your overall results

Some additional resources/references:

- https://time.com/collection/time-100-health-summit-2019/5703540/raj-panjabi-health-care-gap-time-100-health/

- https://www.who.int/hrh/news/2019/community-health-workers-resolution-at-wha/en/

6. PLOS authors have the option to publish the peer review history of their article (what does this mean?). If published, this will include your full peer review and any attached files.

Reviewer #1: No

Reviewer #2: Yes: Rukshan Mehta

---

## [Author Response · Author response to Decision Letter 0]

23 Jan 2020

Reviewer comments and our responses

Journal Requirements:

Our response: Data cannot be shared publicly as participants did not give consent for their transcripts to be shared in the public domain. Data generated from icddr,b’s research can be provided to interested researchers (Recipients) for secondary data analyses upon approval of a Data Licensing Application & Agreement by the icddr,b Data Centre Committee. Interested personnel is recommended to consult this with icddr,b IRB Coordinator Mr. M A Salam Khan (ssalamk@icddrb.org).

Our response: We added captions for supporting information files at the end of our manuscript.

 5. Review Comments to the Author

 Reviewer #1: General Comments

 The manuscript provides a meaningful contribution to our understanding of factors affecting the motivation of Community Based Volunteers who have been incentivized. It is especially important for practitioners in Low and Middle Income Countries, where Community Based Volunteers deliver critical health services to a large proportion of the population mostly living in Rural areas.

 Overall, the authors have attempted to provide clear descriptions and adequate information to understand the topic at hand. Their application of the framework analysis to arrive at the result was sound. However their discussion and conclusion sections need to be redone as they do not adequately capture the results and existing literature in the area of study. Additionally the authors should consider reviewing the grammar of their manuscript as it greatly affects the reading experience.

Our response: We appreciate reviewer efforts in reviewing our manuscript. Below, we responded to reviewer comments and accordingly revised texts in the main manuscript. 

Specific sections

 Abstract

 1. Introduction: Based on the contents of the paper, I don’t think that the paper has evaluated SS performance. It has however evaluated the factors affecting the performance of the Shathya Shebika. I think it would be important to mention that they are voluntary workers but are paid a small incentive in the introduction.

Our response: Revised as suggested in the abstract.

 2. Methods: The first sentence states that the study was qualitative in nature. The authors ought to combine the message in the next sentence to avoid repetition e.g “This was a qualitative study conducted between June 2014- December 2016 as part of a larger evaluation of BRAC’s home fortification programme. Data was generated through in depth interviews, focus group discussions and key informant interviews…….”

Our response: Agreed and revised accordingly in the abstract.

 3. Conclusion: The first sentence does not make sense. The authors can rephrase the sentence to read, “BRAC’s volunteer SS model faces challenges at individual, community, programme and organizational level.”

Our response: Agreed and revised accordingly in the abstract. 

Introduction

 1. Line 88-90: Is the MIYCN ongoing or has it stopped this is quite unclear in the introduction section.

Our response: Revised for better clarification on page….. 

 2. Given that the paper is discussing the SS performance in the MIYCN and factors affecting it, it would be informative for the authors to list some of their functions without necessarily going into detail in Lines 91&92.

Our response: As suggested, revised sentences on lines 96-98.

 3. Line 99: I would suggest using another term instead of profit, as what they are receiving is not a profit rather a proportion of what they make selling the MNP.

Our response: This should be termed ‘profit’ as the SS buy MNP from BRAC at BDT 56 and sell to the caregivers at BDT 75 and SS doing this as part of BRAC business model for community health workers. 

 4. Based on the descriptions provided in the subsequent sections of the paper, the authors should consider the aim/ scope of their paper to be an exploration of the factors affecting performance only and not an analysis of their performance.

Our response: Agreed and revised accordingly on lines 113-116.

 Methods

 1. The authors should consider revising the entire methods section. Despite providing critical information within the section, information that should be provided simultaneously is often disjointed. They should possibly consider the use of COREQ guidelines when writing this section to promote more cohesiveness and enhance the reader’s experience.

Our response: The method section is written based on COREQ guidelines, we used 32 items COREQ checklist and submitted as a supporting file with this manuscript. The COREQ checklist has three domains and seven sections. The description of the method covered almost all items and we mentioned the corresponding page numbers in where you will find the information related to the item. 

 2. The authors state that they conducted interviews and focus group discussions but do not state the sample aside from those which were done with caregivers. This is critical information that they have listed under the results section. They should consider providing it here.

Our response: In a broader sense, we grouped our data collection techniques in two groups: interview and focus group discussion (FGD). We interviewed with a different group of respondents, including SS and caregivers. We mentioned the caregiver interview in the data collection section, on lines 167-169 and 181-184. 

 3. What was the rationale behind interviewing the husbands of SS?

Our response: The SS performance influenced by her husband’s understanding and motivation about her works, after analysing SS interviews, we decided to conduct interviews with SS husband. 

 4. Line 183: Change to FGD rather than GFD and quiet instead of quite.

Our response: Thanks for pointing these errors, it is now corrected on page 9. 

 5. Line 178: Built rather than build.

Our response: Corrected as suggested on page 8. 

 6. Lines 194- 213: This section provides a description of the data that was collected from each of the target populations. The authors could give information on the nature of the data they collected when they are explaining the conceptual framework and state that they developed interview guides based on it. Otherwise, it feels like a repetition to state at the beginning of the section what guided the study then wait and describe it much later on.

Our response: The four main themes of the conceptual framework have been overlaps in the interviews and FGD guidelines. The individual and community-level factors mainly covered in the guideline for the interview with SS and caregivers. The FGD with Shasthya Kormi covered individual, community and programme level factors. The information on factors related to organisation and programme also collected through key informant interviews with BRAC Managers. We added these information lines 203-229 and highlighted the texts in where we mentioned the interview guidelines. 

 7. Lines 213-214: This sentence should appear after Line 184, where the authors describe the actual data collection procedures.

Our response: As suggested, moved this sentence to lines 192-193. 

 Results

 1. It is my understanding that the authors were trying to quantify the themes they came across in the data they analyzed and presented this as Table 2. However I am not sure that this adds value to their findings; For instance they state that in total 180 interviews were conducted, but only 20 participant mention community demand for services from SS as critical to performance. As such does this make it an important finding given that it was mentioned by few people?

Our response: We agreed with reviewer concerns around quantifying qualitative findings, which should not be a feasible approach for qualitative research. Our intention was adding Table 2 not to quantify or generalised the findings at the population level, instead to show the diversity of respondents’ views and opinions on different themes. Despite conducting interviews with 186 respondents, most of the respondents are hardly ally with on any specific theme; presumably, this is a fundamental nature of qualitative research. 

 Discussion

 1. The authors have not provided a robust discussion that situates their work and findings within the literature that exists in the area of factors affecting the performance of community-based volunteers.

Our response: We further expanded the discussion section and reviewed several additional relevant literatures on these discussions. 

 Conclusion

 1. The authors say that one of the limitations of their studies is that they only collected data on the SS who are volunteers. How can this be a limitation if the goal of the study is to evaluate the factors affecting SS performance and not that of paid community health workers?

Our response: This may not be a limitation, but the conclusion we have drawn based on our findings may require further emphasising the issues that we only assess volunteer CHWs. 

 2. Additionally given that their goal was to have an in depth understanding rather than be able to generalize findings. I don’t think they can state that the lack of generalizability of findings is a limitation of their study.

Our response: We agreed and deleted the sentence. 

 Reviewer #2: Overall Comments:

This is a well-designed and comprehensive qualitative evaluation study of an important area. There is a significant need for proper copy editing of the manuscript, as it is hard to follow in several places. Additional efforts to improve organization will aid with reduction of repetition. Issues with grammar and prose are pervasive, kindly review and edit throughout.

Our response: This manuscript has been reviewed and edited by an English speaking native, who is a senior author of this manuscript; therefore, we hope this will now fit with the journal standard.

It is advisable that authors consider organizing their results by the different modalities of data collection, to ensure that they have captured key quotations and themes arising from all stakeholders involved in the process. Alternatively, highlighting more effectively/adding quotations throughout the results section, instead of restricting these to Table 3 will remedy this issue.

Our response: We used a conceptual framework for the performance of CHW to conceptualise the design and analysis of this research. Therefore, we presented our findings based on the framework (Figure 1). As in the framework, there are four main themes (level): individual level, community level, programme level and organisational level factors are associated with the performance of CHW. At the individual level, there are five sub-themes: individual perception, age and education, self-efficacy, and individual career prospect working as SS. Similar way, we presented our findings under the other three main themes. We mentioned in the analysis section on page 11, line 253. We used a range of quotations (almost for every sub-themes) of the study participants, which may increase the results section substantially if we keep them in the texts; therefore, we decided to use Table 3 for presenting them. 

 The role of Shastya Kormis is not clearly defined in the introduction and should be highlighted, particularly in reference to their involvement in provision of supportive supervision.

Our response: The Shasthya Kormis are the paid health staff of BRAC and the primary supervisor of BRAC SS. We revised the texts in the introduction on page 4, line 81-83.

 Consider citing the socioecological framework as your main theoretical framework.

 A recent report by the WHO regarding the role of CHWs in LMICs should be cited, as it substantiates the need for payment and adequate monetary reimbursement to frontline workers. We have seen similar findings from studies conducted in geographically similar regional contexts.

Our response: We agreed, our framework originally developed based on the socioecological framework. In this framework, individuals are attached to a larger socio-ecological system, described the characteristics of individuals and interaction between the individuals and environments that underlie health outcomes. We cited a paper on socioecological framework. We also reviewed several WHO reports/guidelines and cited accordingly. 

 The discussion is rather thin, and can benefit from a review of the literature pertaining to FLW performance and motivation. This is an important and timely issue and the overall findings in the study pertaining to adequate compensation for CHWs should be emphasized and highlighted with support from the existing body of literature in this area.

Our response: We further expanded the discussion section, reviewed several additional relevant literatures, and cited them accordingly.

 Introduction:

 Lines 61-63:

 - Statement is unclear

 - Please highlight the roles and responsibilities of Shastya Kormis, their place in the overall BRAC organizational structure and compensation in relation to Shastya Sebikas

Our response: The original line 61-63 describes an important characteristic of volunteer CHWs. Due to their volunteer nature, they do not have a fixed role or responsibility; rather, responsibilities are mostly flexible. The Shasthya Kormis are the paid health staff of BRAC and the primary supervisor of BRAC SS. We revised these lines for better clarity on page 3, line 60-64 and on page 4, line 81-83.

 Line 106:

 - “The performance of community health workers can be measured by several factors including availability, productivity, competence and responsiveness”

 o Availability of what?

Our response: Availability of CHW (with fewer vacancies or dropout of CHW), specify in the paper on page 5, line 111 

 Materials and methods:

 Lines 119-122:

 - Sentence is unclear, please consider reframing

Our response: Lines 126-27, revised as suggested. 

 Lines 142-147:

 - Can you comment on what point in the overall program this study was conducted? What was the duration between baseline and the evaluation?

Our response: This study was conducted during the first two years of the programme implementation. The duration of home fortification programme was five years from 2014 to 2018, implemented in three phases in three different areas. As we conducted a concurrent evaluation, the evaluation activities have been implemented alongside the programme implementation, and there were a series of evaluation activities that have been implemented as part of this concurrent evaluation. This paper only used qualitative findings of the evaluation, data on the baseline and endline survey (duration was three years) published elsewhere (see reference # 15). 

 Line 159:

 - You mean to say “saturation”, in qualitative work, multiple data collection techniques are used to reach saturation of themes. This also has implications for your sample size.

Our response: Revised as suggested, line 165.

 Lines 159-176:

 - This entire section is very hard to follow.

 - Several points need to be addressed:

 o How many interviews, FGDs and IDIs did you conduct and with which group

 o Kindly reorganize this section by either – type of data collection method i.e. FGD, IDI, KII, or by the target respondent.

Our response: The number of interviews and FGDs we conducted reported at the beginning of the results section under ‘Background characteristics of study participants’. The section has been written sequentially by type of data collection. The second sentence deal described in-depth interviews, then the third sentence about FGDs and fourth and subsequent sentences of this paragraph described key informant interviews. 

 Lines 194-212:

 - Same comments as above, this entire section is unclear and requires reorganization. For ease of organization, consider breaking the section up into sub-headings by type of data collection activity (i.e. FGD, IDI, KII) or target respondent.

Our response: For further clarity of this section, we added a supplementary table (Table S1).

 Data Analysis:

 Line 219:

 - Your cited reference does not match your reference

Our response: Revised the reference, thanks for pointing this. 

 Overall: did you engage in memoing during the data analysis process?

Our response: We consider memoing during transcription and writing interview reports; however, in the final analysis for this paper, we did not engage in memoing.

 Lines 233-234:

 - You mention triangulation of findings, can you clarify if you are referring primarily to the qualitative data or also the quantitative evaluation data? You mention triangulation, however it is unclear based on your results section, what the outcomes of triangulation were.

Our response: We performed triangulation of the data generated through in-depth interviews, FGDs and Key informant interviews. We did not triangulate between qualitative and quantitative (i.e., survey) data. 

 Results:

 Lines 252-262

 - This entire section is very hard to follow, same comment as above, please consider adding sub-headings and reorganizing to improve flow

Our response: We reported the background characteristics of the study participants sequentially as it is in Table 1. At first, we report about SS, then Shasthya Kormi, BRAC’s Programme staff members (i.g., Upazila Manager), and finally, caregivers who are the beneficiaries of above BRAC staff members/health workers). 

Lines 287-289:

 - This sentence is unclear, please rephrase

Our response: We revised as suggested on lines 297-300.

 Lines 300-305:

 - Consider adding a quotation here

Our response: Added a quotation of SS in Table 3 under the sub-theme “Perception about working as an SS influenced performance”. 

 Table 3:

 - Age and education of SS

 o Please add a quotation regarding education

 o Also, clarify whether you are referring to formal education or information/work-related training

Our response: We added a quotation under the sub-themes age and education in Table 3. We are referring to formal education. 

 - Individual level factors:

 o “self-efficacy – hesitation to visit all households”

 § This is unclear, what does self-efficacy have to do with this?

Our response: The self-efficacy refers to an individual's belief in his or her capability to perform behaviours related to produce specific performance attainments. SS with a higher level of self-efficacy would able to overcome any barrier irrespective of their religious and or social identity. The SS with low self-efficacy was hesitated to visit some households/caregivers in her community who are from a different religious group or higher educated than the SS. The low self-efficacy SS does not have enough confidence to motivate a caregiver if the caregiver is coming from a perceived higher level of background. We further clarify it, line 336-338.

 - Programme level factors:

 o “Timely receiving programme….”

 § It is unclear what you mean to say in this section, additionally, the quotation you have cited does not match this factor

Our response: The training department of BRAC is a separate department from the MIYCN programme department. They only organised training for newly recruited SS when they had 20 SS to fulfill a batch. Consequently, when BRAC field office recruited a new SS to fill a vacancy, they could not provide basic training to that SS immediately. In that situation, the SS started working without basic training. When a SS working in the community without training faced more challenges than a trained SS, which influenced their performance in home-fortification implementation. We now revised the section for more clarity, see line 536-554. We also revised the quotation for this theme in Table 3. 

 o “Competing with other programmes…”

 § This is an important factor, you may consider addressing it elsewhere in the manuscript as well. Why is BRAC selling MNPs in the same communities where other programs are ongoing?

Our response: In Bangladesh, BRAC is the only organisation implementing MNP intervention from the very beginning of MNP development (invention) and BRAC scaled up this intervention across the country. Other NGOs in Bangladesh were implementing this as part of piloted interventions and implemented in one or two sub-districts, whereas through MIYCN programme BRAC implemented it in 164 sub-districts. There might need better collaboration between BRAC and other NGOs in order to avoid overlaps. We address this in the discussion as well. 

§ Same comment for section on lines 420-421

 • Why does this overlap exist and in how many communities was this seen?

Our response: During our evaluation we observed overleaps in one sub-district, revised on lines: 558-560

 - Lines 337-349

 o Can you address issues of monetary compensation and/or performance in relation to improved training and performance of CHWs

Our response: We presented findings on monetary incentives and SS performance under programme level factors – Low and uneven distribution of incentives demotivated SS. We did not have data to address the issues of monetary compensation and/or performance to improved training.

 - Lines 378-382

 o Consider adding a quotation here to augment your findings

Our response: A related quotation is available in Table 3 on that argument under sub-theme: Social norms, religious issues, family support influenced the functions of SS. 

 - Lines 479-481

 o Why was a high turnover seen among supervisory level staff? Did your qualitative findings provide any additional insights as to what was driving this phenomenon? What were the implications of this for CHW performance?

Our response: The high turnover seen among supervisory level staff due to low and uneven salary structure for the BRAC staff members at the sub-district level compared to other NGOs working in the regions (Table 3). Unavailability of supervisory-level staff members created a considerable constraint to regular monitoring and giving timely supportive supervision to SS. We revised texts on lines 489-491.

 - Lines 575-578

 o Sentence is unclear – please rephrase

Our response: Revised as suggested on lines 582-585

 - Lines 583-585

 o Sentence is unclear – please rephrase

Our response: Rephrased as suggested on line 590-593

 - I am unclear on what the overall results from your FGDs and KIIs were, can you provide a synthesis of these in the results section.

Our response: As said above, we presented findings based on the conceptual framework for the performance of SS, not based on the data collection techniques i.e., In-depth Interviews, FGDs and KIIs. Table 3 presents the quotations of study participants in different data collection techniques. 

 - Can you briefly comment on the quality of data generated through the different modalities, i.e. in-depth interviews versus FGDs and whether this impacted the data you generated in any way?

Our response: We ensured quality in our qualitative research by following standard procedures. We followed the following steps as recommended for a standard procedure of ideal qualitative research. We describe them in the method section. 

1) We clearly justify our study rationale and aim in the introduction, 

2) We followed rigorous methodological procedures, 

3) We collected data from different participants on the same topics and triangulated to ensure findings robustness, 

4) We ensured interpretative rigour through involving multiple researchers in the analysis and interpretation process, 

5) Ensured reflexivity and evaluative rigour – the researchers who involved in the data collection, analysis and interpretation were from the same community, thus, the awareness by the researchers of the social setting of the research maintained. 

Line 619-621:

 - This sentence is unclear – please rephrase

Our response: Rephrased as suggested on lines 641-643.

 Overall:

 - Can you comment on selection bias and how you addressed this in your sampling strategy?

Our response: As we aimed to generate in-depth, rich and holistic information, we followed purposive sampling, where selection bias is obvious. We were trying to identify the participants with maximum variations and who have the ability to provide rich and in-depth information. A detailed sampling (including inclusion criteria) explained on lines 177-184.

Moreover, in order to ensure quality data collection, we followed standard procedures of qualitative research (mentioned above), involved experienced and skilled qualitative researchers, and implemented the methods perfectly. 

 - Can you comment on any inclusion/exclusion criteria you kept in mind when sampling?

Our response: See above responses.

 - You mention that you interviewed caregivers of beneficiary children, however I am unclear on what your key findings were from these interviews, and how they influenced your overall results

Our response: The caregiver’s findings mainly used to triangulate the study findings provided by SS and other participants. The caregiver’s findings were coming at an individual level and community level. 

 Some additional resources/references:

 - https://time.com/collection/time-100-health-summit-2019/5703540/raj-panjabi-health-care-gap-time-100-health/

 - https://www.who.int/hrh/news/2019/community-health-workers-resolution-at-wha/en/

Our response: Thanks for sharing this, we reviewed them and cited as appropriate.

---

## [Decision Letter · Decision Letter 1]

17 Feb 2020

PONE-D-19-18797R1

Performance of Volunteer Community Health Workers in Implementing Home-fortification Interventions in Bangladesh: A Qualitative Investigation

PLOS ONE

Dear Mr. Sarma,

Thank you for submitting your manuscript to PLOS ONE. After careful consideration, we feel that it has merit but does not fully meet PLOS ONE’s publication criteria as it currently stands. Therefore, we invite you to submit a revised version of the manuscript that addresses the points raised during the review process.

We would appreciate receiving your revised manuscript by Apr 02 2020 11:59PM. To enhance the reproducibility of your results, we recommend that if applicable you deposit your laboratory protocols in protocols.io, where a protocol can be assigned its own identifier (DOI) such that it can be cited independently in the future. For instructions see: http://journals.plos.org/plosone/s/submission-guidelines#loc-laboratory-protocols

We look forward to receiving your revised manuscript.

Kind regards,

Jai K Das

Academic Editor

PLOS ONE

Reviewers' comments:

Reviewer's Responses to Questions

**Comments to the Author**

1. If the authors have adequately addressed your comments raised in a previous round of review and you feel that this manuscript is now acceptable for publication, you may indicate that here to bypass the “Comments to the Author” section, enter your conflict of interest statement in the “Confidential to Editor” section, and submit your "Accept" recommendation.

Reviewer #1: All comments have been addressed

Reviewer #2: All comments have been addressed

2. Is the manuscript technically sound, and do the data support the conclusions?

Reviewer #1: Yes

Reviewer #2: Yes

3. Has the statistical analysis been performed appropriately and rigorously? 

Reviewer #1: N/A

Reviewer #2: Yes

4. Have the authors made all data underlying the findings in their manuscript fully available?

Reviewer #1: Yes

Reviewer #2: Yes

5. Is the manuscript presented in an intelligible fashion and written in standard English?

Reviewer #1: No

Reviewer #2: Yes

6. Review Comments to the Author

Reviewer #1: General Comments

The authors have made a concerted effort to address the comments raised. Additional reviewers for grammar and spelling should be done.

Specific sections

Abstract

1. Line 18 should be receive instead of received.

2. Line 22 consider revising the use of “during the period of June 2014 to December 2016 “ to “between the period of June 2014 to December 2016. “

Introduction

1. Line 93: Consider using between rather than during.

2. Line 110. Revise the structure of the sentence. Are you stating that the performance is measured by various factors or that it is shaped by various factors?

Methods

1. Line 126. The abbreviation for community health workers is wrong.

2. Line 138-143. Authors use of triangulation as described here refers more to synthesis of the data from the quantitative and qualitative evaluations rather than the different qualitative approaches. As such it is not entirely relevant for the description of this particular study. They should consider revising this to reflect that.

3. Lines 203-210. This section should come earlier. Before the paragraph that starts at line 186.

4. Though authors have made some changes to the section, however could the consider breaking it into subsections so that it is somewhat easier to read.

Reviewer #2: Minor edits:

Line 141 - "validity of estimates", what does this mean? You are presenting qualitative findings, unclear on what estimates you are referring to?

Line 184 - change, "caregivers who never used MNP"

Line 257 - "consisted of two"

Line 260 - "responded to all queries"

Line 261 - "asked them to give consent"

Line 282 - "An SS completed on average, 4 years of schooling"

Line 383 - "sometimes community members did not accept..."

Line 631 - "should review existing pay"

7. PLOS authors have the option to publish the peer review history of their article (what does this mean?). If published, this will include your full peer review and any attached files.

Reviewer #1: No

Reviewer #2: No

---

## [Author Response · Author response to Decision Letter 1]

18 Feb 2020

Performance of Volunteer Community Health Workers in Implementing Home-fortification Interventions in Bangladesh: A Qualitative Investigation

Revision round two

Reviewer comments and our responses

Reviewer #1: General Comments

The authors have made a concerted effort to address the comments raised. Additional reviewers for grammar and spelling should be done.

Specific sections

Abstract

1. Line 18 should be receive instead of received.

Response: Revised as suggested

2. Line 22 consider revising the use of “during the period of June 2014 to December 2016 “ to “between the period of June 2014 to December 2016. “

Response: Revised as suggested

Introduction

1. Line 93: Consider using between rather than during.

Response: Revised as suggested

2. Line 110. Revise the structure of the sentence. Are you stating that the performance is measured by various factors or that it is shaped by various factors?

Response: Revised as suggested

Methods

1. Line 126. The abbreviation for community health workers is wrong.

Response: Thanks, corrected it accordingly

2. Line 138-143. Authors use of triangulation as described here refers more to synthesis of the data from the quantitative and qualitative evaluations rather than the different qualitative approaches. As such it is not entirely relevant for the description of this particular study. They should consider revising this to reflect that.

Response: Considering reviewer concerns we have revised this sentence. 

3. Lines 203-210. This section should come earlier. Before the paragraph that starts at line 186.

Response: Agreed and rearranged the section as suggested. 

4. Though authors have made some changes to the section, however could the consider breaking it into subsections so that it is somewhat easier to read.

Response: We split the final paragraph of method section into two paragraphs. 

Reviewer #2: Minor edits:

Line 141 - "validity of estimates", what does this mean? You are presenting qualitative findings, unclear on what estimates you are referring to?

Response: Revised 

Line 184 - change, "caregivers who never used MNP"

Response: Changed as suggested

Line 257 - "consisted of two"

Response: Revised 

Line 260 - "responded to all queries"

Response: Revised

Line 261 - "asked them to give consent"

Response: Revised.

Line 282 - "An SS completed on average, 4 years of schooling"

Response: Revised.

Line 383 - "sometimes community members did not accept..."

Response: Revised.

Line 631 - "should review existing pay"

Response: Revised.

---

## [Editor Report · Decision Letter 2]

9 Mar 2020

Performance of Volunteer Community Health Workers in Implementing Home-fortification Interventions in Bangladesh: A Qualitative Investigation

PONE-D-19-18797R2

Dear Dr. Sarma,

We are pleased to inform you that your manuscript has been judged scientifically suitable for publication and will be formally accepted for publication once it complies with all outstanding technical requirements.

With kind regards,

Jai K Das

Academic Editor

PLOS ONE
---

## [Editor Report · Acceptance letter]

17 Mar 2020

PONE-D-19-18797R2 

Performance of Volunteer Community Health Workers in Implementing Home-fortification Interventions in Bangladesh: A Qualitative Investigation 

Dear Dr. Sarma:

I am pleased to inform you that your manuscript has been deemed suitable for publication in PLOS ONE. Congratulations! Your manuscript is now with our production department. 

With kind regards,

on behalf of

Dr. Jai K Das 

Academic Editor

PLOS ONE